# Q-BASED VARIATIONAL INVERSE REINFORCEMENT LEARNING

## ABSTRACT

The development of safe and beneficial AI requires that systems can learn and act in accordance with human preferences. However, explicitly specifying these preferences by hand is often infeasible. Inverse reinforcement learning (IRL) addresses this challenge by inferring preferences, represented as reward functions, from expert behavior. We introduce Q-based Variational IRL (QVIRL), a novel Bayesian IRL method that recovers a posterior distribution over rewards from expert demonstrations via primarily learning a variational distribution over Q-values. Unlike previous approaches, QVIRL combines scalability with uncertainty quantification, important for safety-critical applications. We demonstrate QVIRL's strong performance in apprenticeship learning across various tasks, including classical control problems and safe navigation in the Safety Gymnasium suite, where the method's uncertainty quantification allows us to produce safer policies.

## 1 INTRODUCTION

Stuart Russell (2019) has suggested three principles to guide the development of beneficial artificial intelligence, from autonomous industrial robots and autonomous vehicles to advanced future systems: AI's only objective is realizing human preferences; AI is initially uncertain about what these preferences are; and the ultimate source of information about human preferences is human behavior. *Apprenticeship learning*[1] via Bayesian *inverse reinforcement learning* (IRL) can be understood as a possible operationalization of these principles: Bayesian IRL starts with a prior distribution over reward functions representing initial uncertainty about human preferences.[2] It then combines this prior with *demonstration* data from a human expert acting approximately optimally with respect to an unknown reward, to produce a posterior distribution over rewards. In apprenticeship learning, this posterior over rewards is then used to produce a policy that should perform well according to the unknown reward function.

Non-Bayesian IRL has been successfully applied in apprenticeship learning in settings including robotics (Kretzschmar et al., 2016; Okal & Arras, 2016; Woodworth et al., 2018; Das et al., 2021; Liu et al., 2022), navigation in Google Maps on a global scale (Barnes et al., 2023), or autonomous driving (Sun et al., 2018; Rosbach et al., 2019; Huang et al., 2022), including on vehicles deployed in real-world heavy traffic (Phan-Minh et al., 2022). However, this body of work that has scaled IRL to real-world settings generally learns only a point-estimate of the reward function, which can be problematic for several reasons: first, the IRL task is generally underspecified – there exist multiple reward functions that equally well explain given behaviour, even in the limit of many demonstration trajectories (Russell, 1998; Cao et al., 2021; Kim et al., 2021; Skalse et al., 2023). Second, further uncertainty is induced by working only with a limited amount of demonstrations. Especially in safety-critical applications, it is desirable to track this uncertainty to subsequently produce policies

---

[1]We use the term apprenticeship learning to mean a subarea of imitation learning that uses IRL as an intermediate step. Imitation learning is thus a broader term including non-IRL techniques for learning a good policy from demonstrations, such as behavioural cloning.

[2]While the reward hypothesis as formulated by Rich Sutton would claim "that all of what we mean by goals and purposes can be well thought of as maximization of the expected value of the cumulative sum of a received scalar signal (reward)." (Sutton, 2004), it is probably not the case that *all* preferences can be meaningfully represented by a reward (Skalse & Abate, 2022), we think that it covers a wide enough array of tasks or preferences to be worth studying, possibly forming basis for later exploration of alternative formalizations.

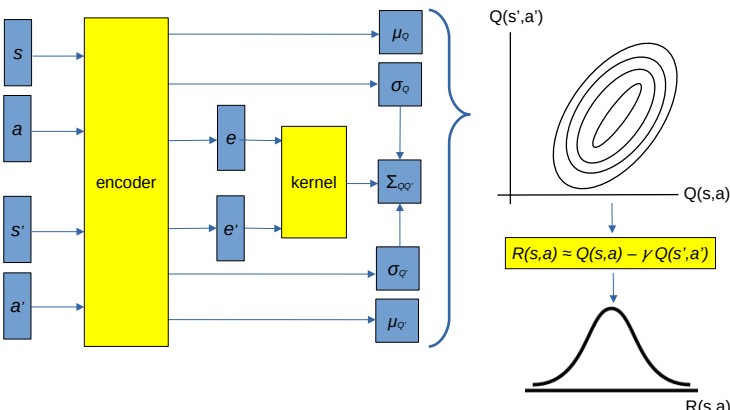

Figure 1: Illustration of a simplified QVIRL architecture. A state-action pair $(s, a)$ and a successor pair $(s', a')$ are fed into an encoder network, which for each pair $(s, a)$ produces the mean $\mu_Q$ and the standard deviation $\sigma_Q$ of the associated Q value $Q(s, a)$, as well as a latent embedding $e$. These embeddings of a set of state-action pairs can be passed to a kernel to produce their correlation matrix and, combined with individual variances, their covariance $\Sigma_{QQ'}$. These together define the joint variational distribution of the Q-values associated with a given set of state-action pairs. This distribution, together with the Bellman equation, can then be used to cheaply deduce the distribution over the corresponding reward.

that are robust with respect to a range of rewards consistent with the observed data, rather than producing a policy optimizing a point-estimated reward that could be wrong.

*Bayesian* IRL addresses these problems head-on by recovering a posterior distribution over reward functions,[3] rather than a point estimate, which can be used to produce policies robust with respect to this posterior uncertainty.[4] However, prior methods for Bayesian IRL (see the Related Work section) are generally applicable only to finite state and action spaces, or continuous spaces with only a handful of dimensions (Ramachandran & Amir, 2007; Mandyam et al., 2023; Bajgar et al., 2024), thus hindering their applicability to real-world settings. Alternatively, other work (Chan & van der Schaar, 2021) sacrifices posterior uncertainty quantification in the interest of scalability.

The contribution of this paper is providing a method that preserves both scalability – in terms of the dimensionality of the state space as well as the amount of demonstrations, which we demonstrate by applying it to higher dimensional settings than any previous article on Bayesian IRL – and full posterior uncertainty estimation, whose usefulness we demonstrate by producing risk-averse policies that can avoid hazards in the Safety Gymnasium. We achieve this by combining variational inference (generally notably more computationally efficient than Markov chain Monte Carlo used in much previous work) and by primarily working in the space of Q-values rather than rewards, which avoids the need to repeatedly solve the forward planning problem – a notorious obstacle in IRL. We see this as a step in scaling Bayesian IRL methods to higher-dimensional settings, broadening the range of tasks that can benefit from the improved robustness that posterior uncertainty quantification can bring, and opening the door to further work on scaling, robust apprenticeship learning, or active IRL.

## 2 TASK

The goal of Bayesian inverse reinforcement learning is recovering a posterior distribution over reward functions based on observing a set of demonstrations $\mathcal{D} = \{(\phi(s_1), a_1), ..., (\phi(s_n), a_n)\}$ from an expert acting in a Markov decision process (MDP) $\mathcal{M} = (\mathcal{S}, \mathcal{A}, p, r, \gamma, t_{\max}, \rho_0)$ where $\mathcal{S}$ and $\mathcal{A}$ are the state and action spaces respectively, $\phi : \mathcal{S} \to \Phi$ is a feature function that maps each state to

---

[3]In general, when talking about Bayesian IRL in this article, we mean methods that model the posterior uncertainty in the learnt reward, thus omitting works doing only maximum-likelihood estimation.

[4]Importantly, it can also be used for active learning to efficiently reduce the posterior uncertainty, but that is not evaluated in this paper.

its representation in a feature space $\Phi$, $p : \mathcal{S} \times \mathcal{A} \to \mathcal{P}(\mathcal{S})$ is the transition function where $\mathcal{P}(\mathcal{S})$ is a set of probability measures over $\mathcal{S}$, $r : \Phi \times \mathcal{A} \to \mathbb{R}$ is an (expected) reward function,[5] $\gamma \in (0, 1)$ is a discount rate, $t_{\max} \in \mathbb{N} \cup \{\infty\}$ is the time horizon, and $\rho_0$ is the initial state distribution. Where there is no risk of confusion, we sometimes write rewards, Q-functions, and policies directly as functions of the state, omitting the feature function, to simplify notation.

In IRL, we know all elements of the MDP except for the reward function and, possibly, the transition function.[6] Instead, we have a model of how the expert policy is linked to the reward and, in the case of Bayesian IRL, we also have a prior distribution over reward functions. Commonly used expert models include: Boltzmann rationality models such as

$$\mathbb{P}[a_i|\phi(s_t)] = \frac{e^{\alpha Q^*(\phi(s_t), a_i)}}{\sum_{a' \in \mathcal{A}} e^{\alpha Q^*(\phi(s_t), a')}} \tag{1}$$

(Ramachandran & Amir, 2007; Chan & van der Schaar, 2021) where $Q^*(s, a)$ is the expected (discounted) return if action $a$ is taken in state $s$ and the optimal policy is subsequently followed, and $\alpha$ is a rationality coefficient; the maximum entropy approach (Ziebart et al., 2008), where the probability of each trajectory is assumed to be proportional to the exponential of the trajectory's return; or sparse behaviour noise models (Zheng et al., 2014), where the expert is assumed to behave rationally except for sparse deviations. Beside these approximately rational models, various models of irrationality can also be considered (Evans et al., 2015). The Bayesian IRL framework is flexible with respect to the choice of expert model (each such model just resulting in a different likelihood function), and can also be extended to the case where the model is not fully known.

In this article, we adopt the Boltzmann rationality model (Eq. 1). We will assume that conditional on the Q values, the actions chosen by the expert are independent, yielding the likelihood

$$p(\mathcal{D}|r) = \prod_{(s_t, a_t) \in \mathcal{D}} \frac{e^{\alpha Q^*(\phi(s_t), a_t)}}{\sum_{a' \in \mathcal{A}} e^{\alpha Q^*(\phi(s_t), a')}} \tag{2}$$

for a discrete action space $\mathcal{A}$ (the expression can readily be adapted to a continuous setting by replacing the sum by an integral). Given this likelihood together with the prior over rewards $p(R)$, we can calculate the posterior using the Bayes Theorem as $p(r|\mathcal{D}) = p(\mathcal{D}|r)p(r)/p(\mathcal{D})$. Note that the full probability of the demonstrations (including the likelihood) would involve also the transition probabilities, but since the same term would appear in both the numerator and the denominator of the posterior, it would cancel out, so we are omitting this probability from the formula for the likelihood.

Generally, we cannot calculate the posterior analytically, so in practice, we need to resort to approximate methods. We propose to use variational inference, in a manner that substantively improves upon the only previous use of variational inference for this problem.

## 2.1 RELATED WORK

Inverse reinforcement learning (IRL) has been introduced by Russell (1998), though essentially the same problem had been formulated and studied before as *inverse optimal control* (Kalman, 1964) (though the two communities have been somewhat separate and using different sets of methods; see Ab Azar et al. (2020) for a comparison). The already vast IRL literature is well reviewed by Arora & Doshi (2021) and Adams et al. (2022) – we will concentrate on approaches closest to ours, i.e. *Bayesian* inverse reinforcement learning methods.

The problem of *Bayesian* IRL was first addressed by Ramachandran & Amir (2007) using Markov chain Monte Carlo (MCMC) sampling. MCMC can produce samples from the true posterior over rewards, but scales poorly to higher dimensions. Furthermore Ramachandran's method needs to solve

---

[5]Our formulation permits the underlying reward to be stochastic. However, our expert model (1) depends on the reward only via the optimal Q-function, which in turn depends only on the expectation of the reward. Thus, the demonstrations only ever give us information about the expectation. Throughout the paper, the learnt reward function can be seen as either modeling a deterministic reward or an expectation of a stochastic one.

[6]The setting without the knowledge of transition dynamics – or other form of access to the environment or its simulator – is sometimes called *strictly batch* (Jarrett & Bica, 2020); our method is applicable in both this setting and the one including an environment simulator, though most of the experiments are run in the former setting following the main baseline method, AVRIL (Chan & van der Schaar, 2021), introduced later.

the forward RL many times. Michini & How (2012) tried to improve the efficiency by focusing only on the most relevant parts of the state-action space. Mandyam et al. (2023) instead tried to reduce the number of times the forward RL problem needs to get solved by solving it a few times with different reward functions and then trying to learn a joint density over rewards and demonstrations using kernel density estimation, but the method can still address only very small problems. Bajgar et al. (2024) significantly reduced the computational burden by performing inference primarily in the space of Q-values, also avoiding the need to repeatedly solve the forward RL problem – a trick we also use. However, the method is still based on MCMC sampling, which is computationally intensive and has trouble scaling to higher-dimensional inference problems.

Chan & van der Schaar (2021) have applied the much more scalable variational inference to the problem. Their approach avoids having to repeatedly solve the forward RL problem by jointly learning a reward encoder (a network producing a mean and variance for the reward in any state) and a Q-network, capturing the current estimate of the expert policy, the two being tied together by the Bellman equation applied as a soft constraint. The Q-network is then used to produce the apprentice policy. However, since the method models only a point estimate of the Q-function, it does not provide any uncertainty estimation for the apprentice policy. Furthermore, since the reward posterior is tied to this point-estimate of the Q-function, its posterior variance is greatly reduced relative to the true Bayesian posterior. Our method, focusing on modelling the uncertainty over Q-values, does not suffer from these flaws.

Other methods (Choi & Kim, 2015; Qiao & Beling, 2011; Wei et al., 2023) have applied elements of Bayesian reasoning in IRL, but do not provide posterior uncertainty quantification instead learning e.g. a MAP estimate, while Balakrishnan et al. (2020) use Bayesian optimization to make (otherwise non-Bayesian) IRL more efficient. Bayesian IRL has also been extended beyond the basic setting studied here e.g. to multiple experts (Choi & Kim, 2012).

## 3 Method

We first provide a high-level summary of our method, *Q-based Variational IRL* (QVIRL), and then proceed by describing each of its components in detail. Our method takes as input a set of demonstrations and a prior over the reward functions, and produces as output (1) a variational approximation to the posterior over Q-functions (which can be understood as encoding the optimal policy under the unknown reward), which can be used to deduce (2) a variational distribution over reward functions.

Classical Bayesian IRL (Ramachandran & Amir, 2007) evaluates the posterior of each hypothetical reward by calculating the associated optimal Q-values, which are then used to evaluate the likelihood (2). However, that can be extremely costly, especially since it needs to be done many times. Thus, we build on an insight used already by Chan & van der Schaar (2021), Garg et al. (2021) and Bajgar et al. (2024): going from the Q-values to the reward is computationally much simpler than going from rewards to the Q-function.

Our model therefore centres on modelling a posterior distribution over Q-values. The Q-values can be used to evaluate the likelihood. And the Bellman equations then allow us to deduce the corresponding rewards in one step, rather than having to solve the whole forward RL problem. By pushing the Q-value distribution through the inverse Bellman operator, we can deduce the corresponding distribution over rewards, which can be used to link the distribution over Q-values to the prior over rewards. We approximate both posteriors by variational distributions, which are optimized using the evidence lower bound (ELBO), as is usual in variational inference.

The architecture is illustrated in Fig. 1. We will now describe each component in turn. For a clearer initial explanation, we will first concentrate on the case of discrete state and action spaces, though most of the equations almost directly transfer to the continuous cases – an exact continuous version of the equations can generally be obtained by appropriately replacing summation by integration. In practice, the integrals need to be approximated by discrete samples, reverting back to the discrete-space equations (where we no longer sum over all states or actions but only over a sample).

## 3.1 Q-VALUE VARIATIONAL POSTERIOR

We model the posterior distribution over Q values using a variational family $q_\theta$, parameterised by a vector of parameters $\theta \in \mathbb{R}^d$, which we can use to evaluate the joint distribution of Q-values for any given set of state-action pairs. Let us highlight why it is important for the Q-value posterior to be able to model the covariances between different Q-values: Adding a constant to all Q-values does not change the associated Boltzmann policy. Thus, *any* demonstration data result in a likelihood that is invariant with respect to adding the same value to all Q-values, usually creating positive correlation amongst them. Also, Q-values of states lying on the same optimal trajectory all depend on rewards of states towards the end of that trajectory, thus again creating a positive correlation.

A range of variational distributions could be used. As a baseline, we work out the details here using a multi-variate Gaussian (in a discrete-space case), or a Gaussian process (in the continuous-state-space case).[7]

The particular architecture we will use for our experiments, and which is outlined in Figure 1 is one based on a neural-network encoder with parameters $\theta$, which for any given state-action pair $s, a$, produces a mean $\mu_Q(s, a; \theta)$, a log standard deviation $\log \sigma_Q(s, a; \theta)$, as well as latent-space embedding $e(s, a; \theta)$, which can then be used as an input to a standard Gaussian process kernel $k$ to calculate the correlation matrix of any set of state-action pairs (by default, we will be using a radial-basis function (RBF) kernel, but our method can accommodate any kernel). Thus, for any collection $(s_1, a_1), \ldots, (s_n, a_n)$ of state-action pairs, the variational posterior will yield a joint multivariate Gaussian distribution with mean and covariance

$$\begin{aligned}
\boldsymbol{\mu}_Q &= \left[ \mu_Q(s_i, a_i; \theta) \right]_{i=1}^n \\
\Sigma_Q &= \left[ \sigma_Q(s_i, a_i; \theta) \sigma_Q(s_j, a_j; \theta) k\big(e(s_i, a_i; \theta), e(s_j, a_j; \theta)\big) \right]_{i,j=1}^n.
\end{aligned} \tag{3}$$

As is usual in variational inference, we will optimize the parameters $\theta$ of this variational distribution using stochastic gradient ascent on the evidence lower bound (ELBO)

$$\text{ELBO}(\theta) = \sum_{s,a \sim \mathcal{D}} [\log p(a|s; \theta)] - \text{KL}(q_R(R; \theta) || p(R)), \tag{4}$$

where the first term is the likelihood (2), $q_R(R; \theta)$ is the implicit variational distribution over the reward which we describe in the next section, and $p(R)$ is the prior distribution over the reward. We now turn to the distribution over rewards and the evaluation of the KL divergence to the prior, and then address the evaluation of the likelihood.

## 3.2 EVALUATING THE REWARD PRIOR AND POSTERIOR

In a discounted MDP, there is a bijection between the expected reward function and the optimal Q function. Thus, the variational posterior that we have just introduced implicitly fully defines a posterior over rewards. In particular, according to the inverse Bellman equation,

$$R(s, a) = Q(s, a) - \gamma \sum_{s' \in \mathcal{S}} p(s'|s, a) \max_{a'} Q(s', a') \tag{5}$$

if the state space is discrete (otherwise, we replace the sum by an integral), which, at the very least, enables us to sample from this distribution by sampling from the joint distribution over Q-values and then using Eq. 5 to get the corresponding reward samples.

Unfortunately, the final term – a maximum of (by default, jointly Gaussian) random variables – cannot be evaluated analytically to yield a closed-form distribution for the reward. Monte Carlo sampling is an option to get an empirical sample from the reward posterior. However, this may make it difficult to efficiently propagate gradient during training.

Noting that the posterior Q-values, treated as random variables, are often correlated (as explained earlier) and that the variance of Q-values of proximate states is often similar, during optimization, we will approximate the maximum of the Q-values by the Q-value with the highest posterior mean

$$R(s, a) \approx Q(s, a) - \gamma \sum_{s' \in \mathcal{S}} p(s'|s, a) Q(s', \arg\max_{a'} \mu_Q(s', a')). \tag{6}$$

---

[7]An extension beyond a strictly Gaussian case can be achieved, for instance, through warping (Snelson et al., 2003). For instance, the sinh-arcsinh transformation (Jones & Pewsey, 2009) can be useful to model skew and kurtosis.

That linearizes the Bellman equation and allows us to approximate the posterior over rewards corresponding to a given posterior over Q-values by a Gaussian with mean and variance given by the following lemma:

**Lemma 1.** *Let $R(s,a)$ be a random variable derived, using Equation 6 from the multivariate normal $\{Q(s,a)|s \in \mathcal{S}, a \in \mathcal{A}\}$ with mean and variance from Equation 3. Then, $R(s,a)$ is normally distributed with mean*

$$\mu_R(s,a) = \mu_Q(s,a) - \gamma \sum_{s' \in \mathcal{S}} p(s'|s,a) \max_{a'} \mu_Q(s',a') \tag{7}$$

*and variance*

$$\sigma_R^2(s,a) = \sigma_Q^2(s,a) - 2\gamma\sigma_Q(s,a) \sum_{s' \in \mathcal{S}} p(s'|s,a)\Sigma_{ss'} + \gamma^2 \sum_{s' \in \mathcal{S}} \sum_{s'' \in \mathcal{S}} p(s'|s,a)p(s''|s,a)\Sigma_{s's''} \tag{8}$$

*where $\hat{\pi}(s) := \arg\max_a \mu_Q(s,a)$ and*

$$\Sigma_{ss'} = \sigma_Q\big(s',\hat{\pi}(s')\big)\sigma_Q\big(s'',\hat{\pi}(s'')\big)k\Big(e\big(s',\hat{\pi}(s')\big), e\big(s'',\hat{\pi}(s'')\big)\Big)$$

*is the covariance between the Q-values of top actions in states $s$ and $s'$.*

The proof can be found in Appendix A.

This implicit reward posterior can be used to calculate the KL divergence to the prior, which is needed to calculate the ELBO (4). If we need an explicit representation of the reward distribution for downstream use, we can also fit a further variational distribution to encode the reward posterior directly. This can be done post-hoc, after the main training of the Q-value posterior is finished.

## 3.3 LIKELIHOOD

Besides the prior, we need to be able to evaluate the likelihood. If the posterior over Q-values is approximated using a variational density $q_\theta$, then the likelihood can be written as $p(\mathcal{D}|\theta) = \prod_{(s,a)\in\mathcal{D}} p(a|s;\theta)$ with

$$p(a|s;\theta) = \int_{Q(s,\cdot)\in\mathbb{R}^{|\mathcal{A}|}} \frac{e^{\alpha Q(s,a)}}{\sum_{a'} e^{\alpha Q(s,a')}} q_\theta(Q(s,\cdot))\, dQ(s,\cdot)$$

where by $Q(s,\cdot)$ we denote the vector of Q-values for each action in state $s$.

The above integral is not known to have a closed-form solution; however, following the mean-field approximation proposed by Lu et al. (2021),[8] we can approximate

$$p(a|s;\theta) \approx \left( \sum_{a'} \exp\left( -\frac{\alpha\big(\mu_Q(s,a)-\mu_Q(s,a')\big)}{\sqrt{1+3\pi^{-2}\alpha^2(\sigma_Q(s,a)^2+\sigma_Q(s,a')^2-2\Sigma_{aa'})}} \right) \right)^{-1} \tag{9}$$

where $\Sigma_{aa'}$ is the covariance between the Q-values of $a$ and $a'$ as calculated via the embeddings and the kernel as $\Sigma_{aa'} := \sigma_Q(s,a)\sigma_Q(s,a')k(e(s,a),e(s,a'))$.

## 3.4 ELBO

If we take a further mean-field approximation for the KL term, we can now calculate the ELBO over a batch of demonstrations $\mathcal{D}_b$ as

$$\text{ELBO}(\theta) \approx \sum_{s,a\in\mathcal{D}_b} \log p(a\,|\,s;\theta) - \sum_{s,a\in\mathcal{S}\times\mathcal{A}} \text{KL}\Big(\mathcal{N}\big(R;\mu_R(s,a;\theta),\sigma_R(s,a;\theta)\big) \,||\, p\big(R(s,a)\big)\Big), \tag{10}$$

where the first term is evaluated using Equation 9 and the second can be evaluated analytically as the KL divergence between two Gaussians. Here we neglect the covariance structure for calculating the KL, but it can be easily accommodated. The parameters $\theta$ of the encoder are then trained by maximizing this ELBO using stochastic gradient ascent.

---

[8]An alternative would be to use stochastic variational inference with the reparameterization trick Kingma & Welling (2013), but in our experiments it did not perform as well not only in terms of training speed, which we expected, but also in terms of the performance of the apprentice agent.

### 3.5 APPRENTICE POLICIES

Once a Q-value model (and possibly a reward model) have been trained, they can be used to produce an apprentice policy for performing the task represented by the learnt reward. In the simplest case, this maximizes either the expected Q-value (which is cheaper if environment dynamics remain the same in deployment) or reward (which may be useful for transfer, but requires running forward RL).

However, since we retain a full posterior distribution, we can also generate conservative apprentice policies that instead optimize for other statistics of the posterior, such as some lower quantile or conditional value at risk (CVaR), choosing actions that should lead to outcomes robust with respect to the epistemic uncertainty encoded by the model. This can also have an effect discouraging the model from entering regions of the state space unseen during training – something that is often addressed using ad-hoc heuristics in literature based on behavioural cloning. We use this conservative apprentice policy in our experiments on the Mujoco and Safety Gymnasium environments, where we use the lower-confidence-bound (LCB) apprentice policy

$$\pi_{\text{LCB}}(s) = \arg\max_a \mu_Q(s, a) - \kappa\sigma_Q(s, a) . \tag{11}$$

where $\kappa$ can be chosen to match a particular quantile.

### 3.6 OTHER SETTINGS

While so far we have described a version of the algorithm for environments with finite, discrete state and action spaces and known dynamics, in Appendix B we describe extensions to continuous spaces, as well as unknown dynamics.

## 4 EXPERIMENTS

We ran experiments in plain apprenticeship learning (where the apprentice aims to maximize expected reward) on 3 classic control environments as well as 3 tasks (each for 3 different demonstration datasets) in the Mujoco physics simulator, which we will now describe in turn. To more explicitly demonstrate the advantages of posterior uncertainty estimation, we also test our method in environments with safety constraints from the Safety Gymnasium suite (Ji et al., 2023a). We provide further details on our experiments in Appendix C.

### 4.1 CLASSIC CONTROL ENVIRONMENTS

First, we evaluate QVIRL in the strictly-batch imitation learning setting against a number of baselines on 3 simulated control environments – Cartpole, Acrobot, and Lunar Lander – which were used for evaluation by both the authors of AVRIL (Chan & van der Schaar, 2021) and ValueWalk (Bajgar et al., 2024) – the two closest prior methods for Bayesian inverse reinforcement learning. Beside ValueWalk and AVRIL, we also include a successful method for strictly-batch imitation learning, Energy-Density Maximization (EDM) (Jarrett & Bica, 2020), as well as simple behavioural cloning (BC) and a random policy.

We reuse the same expert demonstrations as prior work (which were generated using a PPO-trained expert policy) and train the model using 1, 3, 7, 10, or 15 expert trajectories. Similarly to prior work, the learnt (mean) Q-model is then used in an apprentice policy, which is tested from 300 new random initializations of the environment. We repeat each experiment 10 times with a different set of expert trajectories and report mean performance, as well as a confidence interval.

**Results** The results can be seen in Figure 2. Our method, QVIRL, robustly achieves expert-level performance with just a single expert trajectory for both Acrobot and Cartpole, unlike any of the other methods. On Lunar Lander, our method outperforms AVRIL – the closest IRL method – and performs comparably to BC, EDM, and ValueWalk. However, compared to ValueWalk – the prior state-of-the-art Bayesian IRL method on these tasks – QVIRL's training time is under a minute (similarly to AVRIL) while in contrast the MCMC-based ValueWalk takes hours and thus our method presents a much faster alternative to the exact inference of ValueWalk, making QVIRL useful in cases where ValueWalk's computational cost cannot be afforded.

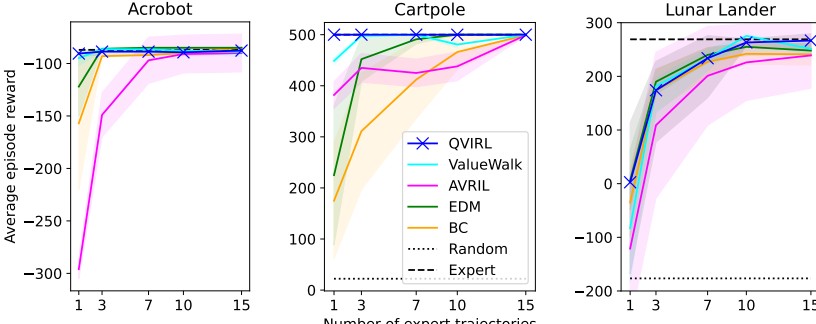

Figure 2: The performance of an apprentice agent for QVIRL and 4 baseline methods. For each number of demonstration trajectories, we report a mean episode return across 10 runs of the algorithm with a different set of expert demonstrations (drawn randomly from a pool of 900). The shaded region for QVIRL and ValueWalk indicates a 90% confidence interval on the value of the mean calculated using a nested bootstrap (resampling the trained models in the outer loop and the episode rewards in the inner loop). For the left two plots, the performance of QVIRL starts at approximately expert level already for a single expert demonstration trajectory, so more challenging settings may be needed for evaluation (and furthermore the horizontal expert performance line is mostly hidden behind the other curves). In the left plot, the random performance is -500, but the plot was cropped to make the differences between the methods easier to see.

**Running times**   A single run of our experiment on the above environments took on average 8.4 minutes of wall time on a single 8-core CPU, similarly to AVRIL's 5.5 minutes. By comparison, the MCMC-based ValueWalk (which has claimed to be already significantly faster than prior work based on MCMC) takes 21 hours to achieve good mixing.

## 4.2 Mujoco

To illustrate the method's performance in a more challenging, continuous-action setting, we also evaluate on 3 environments using the Mujoco simulator (Todorov et al., 2012), namely Hopper, Half-cheetah, and Walker2D, as was done in recent work on continous-action imitation learning (Lyu et al., 2024), whose experimental protocol we also follow. That means we train the model based on a single expert trajectory drawn from the D4RL dataset (Fu et al., 2021) while using other trajectories from the dataset as auxiliary data to approximate the environment dynamics (and thus compare the Q-value variational posterior to the reward prior at various points of the state-action space). Note that these auxiliary trajectories are not assumed to come from an expert and could be generated using an arbitrary (even) random policy. We then evaluate of the apprentice on the environment, reporting the normalized test score following Lyu et al. (2024). We adapt QVIRL to the continuous-action setting by sampling 100 random actions in each step and then use the discrete-action equations given in the previous section. We compare the performance of our algorithm to that of recent algorithms for offline imitation learning: DemoDICE (Kim et al., 2022), SMODICE (Ma et al., 2022), PWIL (Dadashi et al., 2021), and SEABO (Lyu et al., 2024), using IQL (Kostrikov et al., 2022) as the basis for PWIL and SEABO.

**Results**   The results are shown in Table 1. It shows that even on the setting of plainly maximizing the expected return, for which all other methods were optimized, QVIRL produces results competitive with the state of the art and thus constitutes a competitive apprenticeship learning method. That said, we think its main advantage lies in being able to produce conservative policies as we demonstrate on the Safety Gymnasium environment.

## 4.3 Safety Gymnasium

Finally, to demonstrate how we can usefully leverage the posterior uncertainty esimation, we evaluated our model on 3 tasks from the Safety Gymnasium suite (Ji et al., 2023a). Each task has a primary goal (such as reaching a target location) associated with a primary reward as well as a set of

Table 1: **Imitation learning on Mujoco**. Beyond safety settings with risk-averse policies, QVIRL is also a competitive method on general imitation learning, which we illustrate by results on Mujoco with D4RL demonstrations. We use IQL as the base algorithm for SEABO and PWIL. PWIL-action means that we concatenate state and action to compute rewards in PWIL. We report the mean performance at the final 10 episodes of evaluation for each algorithm, $\pm$ standard deviation.

| Task Name | DemoDICE | SMODICE | PWIL-action | SEABO | QVIRL |
|---|---|---|---|---|---|
| halfcheetah-medium | 42.5±1.7 | 41.7±1.0 | 44.4±0.2 | **44.8**±0.3 | 43.2±1.6 |
| hopper-medium | 55.1±3.3 | 56.3±2.3 | 60.4±1.8 | **80.9**±3.2 | 79.8±2.3 |
| walker2d-medium | 73.4±2.6 | 13.3±9.2 | 72.6±6.3 | 80.9±0.6 | **83.8**±0.4 |
| halfcheetah-medium-replay | 38.1±2.7 | 38.7±2.4 | 42.6±0.5 | 42.3±0.1 | **44.8**±0.3 |
| hopper-medium-replay | 39.0±15.4 | 44.3±19.7 | 94.0±7.0 | 92.7±2.9 | **95.8**±1.4 |
| walker2d-medium-replay | 52.2±13.1 | 44.6±23.4 | 41.9±6.0 | 74.0±2.7 | **79.2**±2.8 |
| halfcheetah-medium-expert | 85.8±5.7 | 87.9±5.8 | 89.5±3.6 | 89.3±2.5 | **92.1**±3.1 |
| hopper-medium-expert | 92.3±14.2 | 76.0±8.6 | 70.9±35.1 | **97.5**±5.8 | 89.9±5.2 |
| walker2d-medium-expert | 106.9±1.9 | 47.8±31.1 | 109.8±0.2 | 110.9±0.2 | **112.5**±0.3 |
| Total Score | 585.3 | 450.6 | 626.1 | 713.3 | **721.1** |

Table 2: **Safe imitation on the Safety Gymnasium**. The numbers in the table are the average episode returns with respect to the primary reward ($J_R$, in green) and the safety cost ($J_c$, in red). QVIRL-mean and QVIRL-RA stand for QVIRL-based apprentice policies optimizing for the mean or a (**r**isk-**a**verse) 0.1-quantile of the posterior Q-function distribution respectively. Standard errors can be found in Table 3 in the Appendix.

| Task Name | BC | | IQ | | AVRIL | | AVRIL-online | | QVIRL-mean | | QVIRL-ra | |
|---|---|---|---|---|---|---|---|---|---|---|---|---|
| | $J_R$ | $J_C$ | $J_R$ | $J_C$ | $J_R$ | $J_C$ | $J_R$ | $J_C$ | $J_R$ | $J_C$ | $J_R$ | $J_C$ |
| CarGoal | 0.8 | 21.8 | 7.3 | 44.1 | 1.8 | 62.3 | 3.4 | 22. | 6.2 | 24.6 | 9.5 | 15.0 |
| PointPush | 5.2 | 32.0 | 10.2 | 41.8 | 8.3 | 59.2 | 9.4 | 37.3 | 9.3 | 18.8 | 7.8 | 0.6 |
| RacecarCircle | 16.3 | 78.4 | 19.6 | 39.7 | 18.4 | 44.8 | 17.2 | 32.4 | 22.6 | 31.9 | 21.1 | 3.2 |

safety constraints (such as avoiding hazardous obstacles or leaving the boundaries of a safe region) whose violation produces a safety cost. We provide each method with 10 demonstration trajectories that never violate the constraints. These experiments were run in the online setting, where additional auxilliary trajectories for evaluating the KL term are sampled from the apprentice policy throughout the training. This setting is used for QVIRL, Inverse Soft Q-Learning (IQ; Garg et al. 2021), and AVRIL-online (an online extension of AVRIL; see Appendix D.3). BC and original AVRIL were run in the offline setting for which the algorithms were designed (we are including BC as a standard simplest baseline, and offline AVRIL as the closest prior method). To test it potential for producing risk-averse policies, we used QVIRL to either choose the action with the highest *mean* Q value, or the action with maximum 0.1-quantile Q-value, according to the learnt posterior over Q-values. For each method we evaluated both the return with respect to the primary reward, and the return with respect to the safety cost, where 0 represents no constraint violation, and higher values represent increasingly frequent safety violations.

Note that our method does not internally separate the cost and the reward, but its internal representation can represent unsafe (high-cost) regions with negative rewards, thus still resulting in apprentice policies that avoid these regions. This makes our setting distinct from the constrained-MDP setting Altman (1999), where the primary reward should be maximized subject to a *constraint* on the cost.

**Results** Results are shown in Table 2. The mean version of QVIRL outperforms BC, and AVRIL in terms of the primary reward, while being comparable to IQ. However, the behaviour leads to frequent safety violations. Our posterior uncertainty quantification allows us to also produce a risk-averse policy, and the results show that although it can lead to a slight reduction in the primary reward, the amount of safety violations is significantly reduced, as seen by a lower safety cost on all studied tasks. While Chan & van der Schaar (2021) present AVRIL as a Bayesian IRL method, they learn only a point estimate of the Q-function and use that as a basis for the apprentice policy. This does not straightforwardly allow producing a risk-averse policy.

## 5 DISCUSSION AND CONCLUSION

### 5.1 LIMITATIONS AND ASSUMPTIONS

**First-person demonstrations**  This article assumes that the demonstrations come from an expert acting from the same perspective as the apprentice agent. This may be true in settings such as autonomous driving, where the AI apprentice should be trying to get to destination fast but safely, similarly to what a human driver (from which the demonstration data may come) would be trying to achieve. However, in general, a human and an AI assistant may be acting in the same environment together with the common goal of fulfilling the human's preferences. If the human is observed drinking coffee, our naive IRL framing would teach the AI assistant to also drink coffee. Instead, in such situations, we would want the AI to assist the human in fulfilling their preferences – e.g. brew and bring them a mug of coffee. Such a setting has been formalized as *cooperative inverse reinforcement learning* (Hadfield-Menell et al., 2016) or assistance games (Shah et al., 2020). However, solving those is generally more demanding than plain IRL, so we see Bayesian IRL as a natural step toward applying similar methods in assistance games.

**Expert model**  The method as presented is assuming that the expert is behaving Boltzmann-rationally (per Eq. 1) with a known rationality coefficient $\alpha$. Although this model is known to be relatively robust, real human experts are unlikely to behave exactly according to this model, which can constitute model misspecification. A principled approach to tackle this would be to include uncertainty over reward models, or at the very least, to use Bayesian inference to infer also the rationality parameter, both of which can be readily included in our method, though we have not tested it experimentally so far. More fundamentally, human preferences may be evolving over time and even be influenced by the actions of an AI system (Carroll et al., 2024), all of which would need to be taken into account when robustly aligning AI systems with real users. Still, we hope the probabilistic modelling tools examined in this paper could be used in that endeavour.

**Gaussian variational distribution**  While the general framework presented in the paper is compatible with any variational distribution family, the version empirically tested in this paper was using a Gaussian variational distribution. This means that the method assumes that the two posterior distributions (one over Q-values and one over rewards) are well approximated by this Gaussian (e.g., is unimodal and symmetric), which may not hold in general. The variational distribution needs to be adapted to correspond reasonably well to the true posterior. One way of testing the goodness of fit would be to first use one of MCMC-based methods, such as those by Ramachandran & Amir (2007) or Bajgar et al. (2024), on a smaller instance of a task to produce samples from the true posterior, and only then scaling up in a way that preserves the general shape of the distribution.

**Naïve sampling of continuous actions**  The extension of the method to continuous action has so far used naïve random sampling. This works for low-dimensional action spaces (that is, those comprising a few input signals), but it suffers from the curse of dimensionality, which may prevent this approach from scaling to higher dimensional settings such as robots with many actuators. In such cases, relevant actions – e.g. meaningful movement of the robot, such as a step forward or grabbing an object – may occupy only a small manifold within the many-dimensional action space. In that case, uniform sampling may mostly produce completely irrelevant actions, such as random jerks in a robot. Instead of uniform sampling, one could train a generative policy model that would propose only a few relevant actions which could then be evaluated using the Boltzmann rationality model, with some similarity to actor-critic methods, which we leave for future work.

### 5.2 CONCLUSIONS

We have introduced the first method for Bayesian inverse reinforcement learning that both scales beyond small environments, and at the same time preserves the desirable posterior uncertainty quantification of Bayesian inference. The algorithm performs well not only against baselines within Bayesian IRL, but is competititve also with state-of-the-art algorithms in offline imitation learning. Thus it presents an important step forward in scaling Bayesian inverse reinforcement learning, with its desirable uncertainty quantification properties, into higher-dimensional settings.

## REPRODUCIBILITY

We provide additional details for reproducing our experiments in the Appendix C. We will release full code, including the exact scripts used to run our experiments and Jupyter notebooks used to analyse the results, on Github once the anonymity requirement is lifted. We are also happy to provide a link to the code upon request during the Openreview discussion.

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

## A  PROOF OF LEMMA 1

*Proof.* We use the fact that if a random vector $X$ in $\mathbb{R}^n$ has multivariate normal distribution with mean $\mu$ and covariance $\Sigma$, and $A \in \mathbb{R}^{m \times n}$ is a matrix, then $Y = AX$ also has a multivariate normal distribution with mean $A\mu$ and covariance $A\Sigma A^T$. Note that Equation 6 can be re-written in vector form as

$$R(s, a) = A\mathbf{Q} \,, \tag{12}$$

where

$$\mathbf{Q} = \left( Q(s, a); \; Q(s', \arg\max_{a'} \mu_Q(s', a'))\big|_{s' \in \mathcal{S}} \right)^\top \tag{13}$$

and

$$A = \left( 1, \; -\gamma p(s_1|s, a), \; \ldots, \; -\gamma p(s_n|s, a) \right). \tag{14}$$

The lemma directly follows by applying the above identity relating multivariate Gaussians. □

## B  QVIRL WITH CONTINUOUS SPACES AND UNKNOWN DYNAMICS

Section 3 presented mainly a version of the algorithm for environments with finite, discrete state and action spaces with a known model of the environment. However, the key ideas are applicable across a range of settings, including those with unknown dynamics and with continuous state and action spaces. Here, we outline the two versions that were used for our experiments and which together cover both continuous spaces and unknown dynamics.

### B.1  STRICTLY-BATCH SETTING

In our experiments on Cartpole, Acrobot, and Lunar Lander, we follow the setting of the main baseline algorithm, AVRIL, that is the setting of strictly-batch offline imitation learning (Jarrett & Bica, 2020) where we have access to only the expert demonstrations, and do not have access to the environment or its dynamics during training. These environments (and the baseline algorithm) feature a continuous state space and discrete actions, so let us describe how to adapt QVIRL to that setting.

This can be done in at least two ways: firstly, we could try learning a posterior Bayesian model of the environment together with the reward. Secondly, we can replace sampling over transitions which is needed in Equation 6, by the empirical transitions observed in the expert demonstrations. To make comparison between the algorithms closer, we follow Chan & van der Schaar (2021) in the second approach.

In that case, Equation 6 becomes

$$R(s, a) \approx Q(s, a) - \gamma Q(s', \arg\max_{a'} \mu_Q(s', a')) \tag{15}$$

for $s, a, s', a' \in \mathcal{D}$ – two consecutive transitions from the demonstration data. The KL divergence to the prior is then evaluated only on these demonstration points. If we adapt Equations 7 and 8 accordingly, we have a version of the algorithm applicable to this continuous-state strictly-batch offline setting as we have used it for our experiments.

However, note that in this setting, even the KL term is evaluated only on the expert trajectories. Thus, away from these trajectories, the Q-function does not receive any training signal and becomes unreliable. We think this setting can erase an important part of the advantages that IRL methods can

have over behavioural cloning, which is their ability to leverage the environment dynamics to infer good policy even in states unvisited by the expert. We think our method is more useful in settings where some additional source of information about the environment dynamics is present, such as the ones described in the next subsection.

### B.2 UNLABELLED DATA (INCLUDING ONLINE DATA)

The second evaluation setting we considered following Lyu et al. (2024) contains additional unlabelled transition data, collected using an unknown, possibly random, policy. In that case we can just reuse Equation 15 but instead of evaluating only across demonstrations, we can evaluate the equation also across those unlabelled trajectories, which can help the model generalize to parts of the state space not covered by the demonstrations. Following the baseline algorithm, this is the setting we use for experiments in Section 4.2.

Furthermore, if we have access to a simulator of the environment (or have access to the actual environment itself and it is safe to explore using arbitrary policies), we can also periodically collect trajectories during the IRL training process using the apprentice policy corresponding to current approximate posterior distribution and its random perturbations. Using such auxiliary policies from the apprentice policy has the advantage of improving the consistency of the Q values in regions of the state-action space where it matters the most, since they are likely to be visited by the apprentice policy upon deployment. Adding random perturbations further extends this to the neighbourhood of such apprentice trajectories.

An important effect that this has is that in regions far from the expert data, this makes the Q-value posterior revert to the implicit prior corresponding to the (explicitly provided) reward prior. In particular, this generally results in the Q-value posterior variance being higher in those regions. A risk-averse apprentice policy that penalizes high-uncertainty states and actions will then tend to avoid such regions unvisited by the expert, recovering what is often added as a heuristic in other methods (Brantley et al., 2019; Reddy et al., 2019; Lyu et al., 2024).

### B.3 CONTINUOUS ACTIONS

To adapt the algorithm to continuous-action environments, we simply subsample a set of random contrastive actions from the action space for each training example at each step, and replace the discrete action space by this sampled set of action in all equations. We use 100 sampled actions. Note that this can become inefficient for higher-dimensional action spaces and could be improved by using a trained policy network to propose good candidate actions as is done e.g. by Garg et al. (2021).

## C EXPERIMENT DETAILS

### C.1 DEMONSTRATIONS

For classic control environments, we used the demonstrations provided by Chan & van der Schaar (2021) while for the Mujoco experiments, we used demonstrations from D4RL (Fu et al., 2021) in the same way as Lyu et al. (2024).

On Safety Gymnasium, we first trained an expert to maximize the primary reward while avoiding safety violations using Constrained Policy Optimization (Achiam et al., 2017) as implemented in Omnisafe (Ji et al., 2023b). We used the default settings in Omnisafe except for reducing the cost threshold to 5 (reducing to 0 tended to hinder learning of a reasonable policy). We then used this trained policy to produce the demonstrations, discarding those that had safety cost higher than the threshold.

### C.2 IRL TRAINING

In our experiments we train by stochastic gradient ascent using automated differentiation in PyTorch with the Adam optimizer and a learning rate of $0.001$ (using the default values for other parameters). We use a Boltzmann coefficient of $\alpha = 1$, and a discount rate $\gamma = 0.99$. Each run was run for about

Table 3: **Safe imitation on the Safety Gymnasium. Results with error bars.** The numbers in the table are the average episode returns with respect to the primary reward ($J_R$, in green) and the safety cost ($J_c$, in red). QVIRL-mean and QVIRL-RA stand for QVIRL-based apprentice policies optimizing for the mean or a (**r**isk-**a**verse) 0.1-quantile of the posterior Q-function ditribution respectively. The number after $\pm$ is the standard error of the mean.

| Task | | BC | IQ | AVRIL | AVRIL-online | QVIRL-mean | QVIRL-ra |
|---|---|---|---|---|---|---|---|
| CarGoal | $J_R$ | $0.8 \pm 0.6$ | $7.3 \pm 3.2$ | $1.8 \pm 0.9$ | $3.4 \pm 1.9$ | $6.2 \pm 2.1$ | $9.5 \pm 7.0$ |
| | $J_C$ | $21.8 \pm 11.0$ | $44.1 \pm 14.5$ | $62.3 \pm 19.8$ | $22.7 \pm 14.4$ | $24.6 \pm 7.5$ | $15.0 \pm 4.0$ |
| PointPush | $J_R$ | $5.2 \pm 4.3$ | $10.2 \pm 5.1$ | $8.3 \pm 3.7$ | $9.4 \pm 5.1$ | $9.3 \pm 4.0$ | $7.8 \pm 4.6$ |
| | $J_C$ | $32.0 \pm 15.4$ | $41.8 \pm 17.3$ | $59.2 \pm 23.0$ | $37.3 \pm 19.8$ | $18.8 \pm 6.9$ | $0.6 \pm 0.2$ |
| RacecarCircle | $J_R$ | $16.3 \pm 14.1$ | $19.6 \pm 8.7$ | $18.4 \pm 8.0$ | $17.2 \pm 9.1$ | $22.6 \pm 8.2$ | $21.1 \pm 14.2$ |
| | $J_C$ | $78.4 \pm 41.5$ | $39.7 \pm 13.6$ | $44.8 \pm 12.9$ | $32.4 \pm 14.1$ | $31.9 \pm 11.6$ | $3.2 \pm 0.8$ |

200000 iterations with a batch size of 64. Where a neural network encoder is used, we use a multi-layer perceptron with two hidden layers of 64 units and an ELU activation function for the classic control experiments, whereas on Mujoco and Safety Gymnasium we use two layers of 128 units and a ReLU activation. For classic control and Mujoco these choices were made to match prior work evaluated on the tasks. For the Safety Gymnasium, we just kept the setting from Mujoco. Embedding size of 4 is used in our experiments. We also used weight decay of $0.001$. We tuned only the learning rate and weigh decay, which were chosen by trial and error before the whole set of experiments whose results are reported was run. At most 5 tries were made for each hyperparameter.

Each experiment was run 10 times with a different random seed, and the resulting apprentice policy was then tested across 100 test episodes on the environment. We report mean across these experiments.

Baseline results reported were taken from Chan & van der Schaar (2021) and Bajgar et al. (2024) for Section 4.1.

### C.3 PRIOR DISTRIBUTION

As the reward prior, we used a zero-mean independent Gaussian for all state-action pairs. In most cases, we used a standard deviation of 1. In the case of the Lunar Lander environment, we used a standard deviation of 5.

In continuous environments, a use of a Gaussian-process prior would be more principled to encode smoothness of rewards. However, this introduces additional hyperparameters and may open the experimental protocol to criticism of gaining advantage due to an author-selected informative prior. When deploying the method in practice, the prior should of course be chosen to represent all available knowledge as accurately as possible.

### C.4 EXTRA RESULT DETAILS

To improve readability, we have removed the statistical uncertainty information from Table 2 with the Safety Gymansium results in the main text. We give the full table including the standard error.

## D ADDITIONAL EXPERIMENTS

### D.1 ILLUSTRATIVE GRIDWORLD EXPERIMENT WITH A SINGLE UNKNOWN STATE

For a human readable and easily interpretable illustration of what our method is doing, we ran our method on the simple gridworld in Figure 3 (left) where the top right state is terminal and we use a discount factor of $\gamma = 0.8$. We assume the learner knows all the rewards (which we model as a tight normal prior with std 0.1 centred at the true value) except for the right middle tile, where it has a normal prior with mean -1 and std of 10. We let it observe a single expert demonstration shown in the middle subplot. We are particularly interested in how an apprentice agent would behave in the bottom right cell – would it be worth cutting through the unknown reward tile (to get to the goal

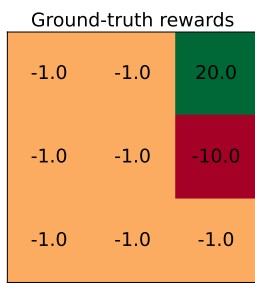  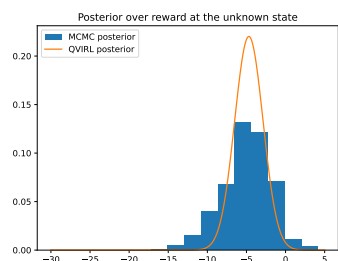

Figure 3: The ground truth rewards, demonstrations, and the posterior over the unknown right-middle reward recovered by QVIRL and Valuewalk in the illustrative gridworld.

faster), or should it go around? The expert demonstration would be consistent with either being optimal.

The right subplot shows the reward posterior recovered by QVIRL (as well as an MCMC reference posterior recovered using ValueWalk, which produces samples from the true posterior). We find that if we use value iteration to find the apprentice policy maximizing the return with respect to the mean reward according to this posterior, the apprentice chooses to go through the unknown-reward tile (with Q-value of 8.0 for going up vs a Q-value of 5.2 of going to the left). On the other hand, if we solve for maximizing the mean minus 2 std of the posterior, representing a risk-averse policy, the Q-value of going up drops to 2.7 and the apprentice would choose to go around.

This illustrates how modelling posterior uncertainty allows us to behave in a risk-averse manner. Furthermore, the plot shows that the reward posterior deduced from the Q-value posterior fits well with the reference reward posterior.

We assume known deterministic dynamics, expert rationality coefficient $\alpha = 1.$, and $\gamma = 0.8$.

### D.2   3X3 GRIDWORLD AND A COMPARISON TO AVRIL

We also tested on a 3x3 gridworld shown in Figure 4, this time treating all rewards (1 per state) as unknown. We assume an independent normal prior for the reward in every state with mean 0 and standard deviation of 66. There is also noise in the environment dynamics, so for any action, there is a 0.1 probability of slipping and moving in a random direction or staying in place (with uniform probability between the 5 options). The top right state yields a reward of 100, while the middle top tile represents a hazardous obstacle with a reward of -30.

We provide the algorithms with 5 trajectories, each starting in the top left corner and heading to the goal via the middle tile while avoiding the obstacle. The expert once slipped to the bottom right tile placing one demonstration step there.

We ran QVIRL and a dynamics-aware version of AVRIL (described below) using these demonstrations, as well as an MCMC method, ValueWalk (Bajgar et al., 2024) that produces samples from the true posterior.

We can see that the QVIRL variational posterior seems to approximate the true posterior well, while the AVRIL point estimate remains very close to zero (which, however, still produces action predictions consistent with the data). The fit of the derived posterior over the rewards is less good – some of the posteriors are visibly further from Gaussian (many exhibiting negative skew) than the value distributions. Still, the derived Gaussian distribution over rewards deduced from QVIRL still roughly matches the means and variances of the MCMC samples.

On the other hand, the posterior over rewards recovered by AVRIL remains centred near zero in almost all cases. Also, in 5 of the 9 states, the standard deviation of the posterior collapses to below 0.2, where it should be between 40 and 64.

Furthermore, AVRIL contains an important hyperparameter $\lambda$ regulating the strength of the constraint on consistency between the Q values and the reward posterior, and we found the results to be extremely sensitive to its value. In the above experiments we used a value of $\lambda = 0.2$, however,

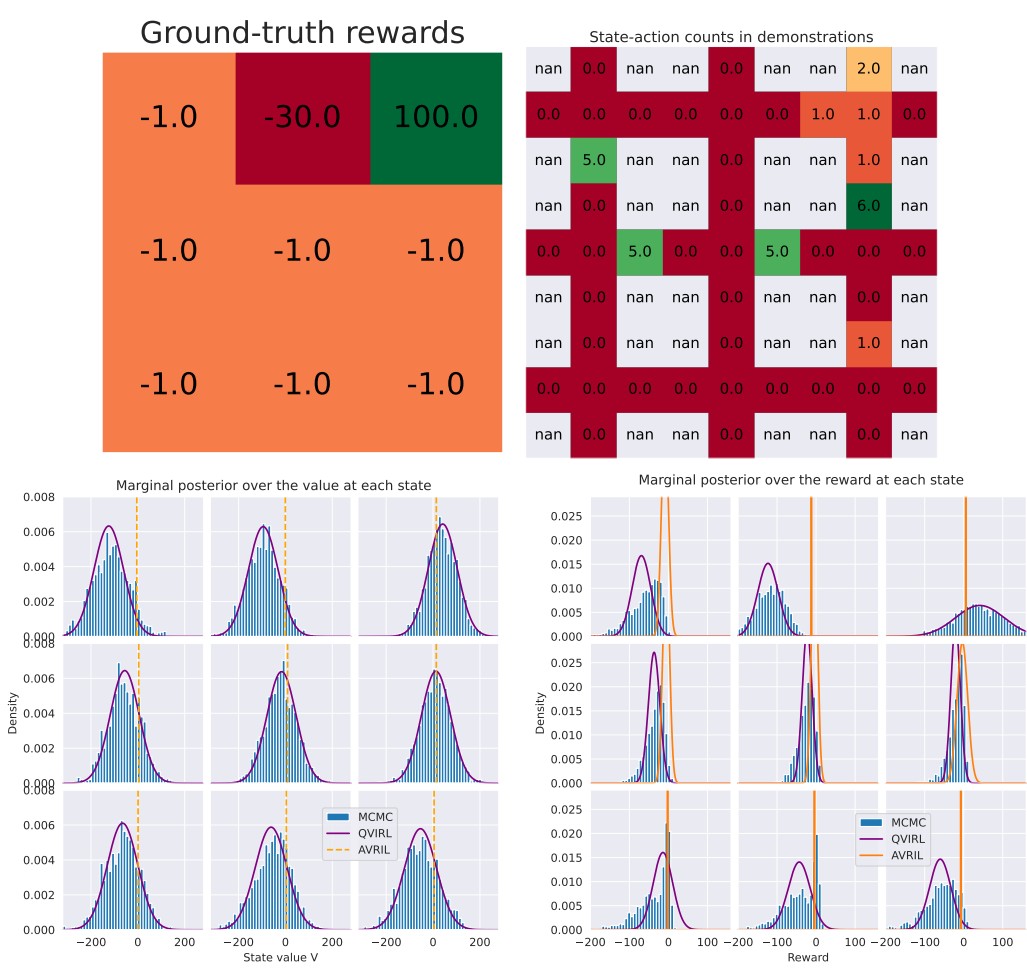

Figure 4: Top left: Ground truth rewards for a 3x3 gridworld. The top left state is the initial state. Top right state is terminal. Top right: State-action counts in the demonstration data. Bottom left: pdf of the marginal posterior over state values recovered by QVIRL, a histogram of 2000 samples from the true posterior produced by an MCMC method (ValueWalk), and a horizontal line indicating the point estimate recovered by AVRIL. Bottom right: pdfs of the variational posterior over reward recovered by QVIRL and AVRIL and the corresponding histogram.

increasing it to $0.5$ makes the variance of all states collapse to near zero (std¡0.001 for 8 states out of nine), while decreasing to $0.1$ essentially reverts the posterior to the prior. The AVRIL paper gives no indication of how to pick a good value for $\lambda$.

### D.3 DYNAMICS-AWARE AVRIL

AVRIL was proposed in a *strictly batch* setting, not using the environment dynamics. In the above case, this would lead to AVRIL never updating the posterior over the reward associated with the obstacle, since this state is never visited and AVRIL updates only states that appear in the demonstrations. Here we assume the knowledge of dynamics, so for fair comparison we introduce a dynamics-aware version of AVRIL.

The main change is that instead of evaluating both the TD term and the KL divergence between the reward variational posterior and the prior only on the expert trajectories, we evaluate it across all state-action pairs, i.e. for each state-action pair $s, a$, we calculate

$$R(s, a) = Q_\theta(s, a) - \gamma \mathbb{E}_{s'|s,a} V_\theta(s'),$$

where $V_\theta(s') = \max_{a'} Q_\theta(s', a')$. Then the TD soft constraint is calculated as

$$\sum_{s,a \in \mathcal{S} \times \mathcal{A}} q_\phi(R(s, a)). \tag{16}$$

. Similarly, we evaluate the KL divergence between $q_\phi$ and the prior $p_r$ across all states and actions rather than just on the trajectories.

### D.4 LIMITATIONS OF AVRIL

Since our method, QVIRL, may seem to resemble AVRIL, the closest baseline, we would like to re-emphasize some important limitations of AVRIL that our method does not suffer from:

- AVRIL does not provide uncertainty estimates over the Q-values and thus over the optimal policy. This does not permit one to easily extract risk averse policies, where we can, which can improve safety as demonstrated in the SafetyGymnasium experiments.

- AVRIL's posterior over rewards does not seem to track well the true posterior over rewards even on a simple gridworld. QVIRL's fit is better.

- the posterior variance over rewards is extremely sensitive to AVRIL's hyperparameter $\lambda$ that regulates the strength of the consistence constraint between the reward variational distribution and the Q values and a narrow range of values can make the posterior either collapse to Dirac delta, or revert to the prior. The paper gives no indication on how to pick the value. Our QVIRL has no such hyperparameter.

- the AVRIL paper presents a "strictly batch setting" where only the expert demonstrations are used to estimate environment transitions, and thus evaluate the closeness of the posterior to the prior and the consistency between the Q values and the reward distribution. We think this largely gives up a crucial advantage of IRL over behavioural cloning – leveraging the environment dynamics to make inference about rewards away from the expert trajectories, enabling better generalization. Our paper presents variants of the algorithm also for the case of known environment dynamics, or an online setting where we have access to a simulator or the environment itself.

## E    APPROXIMATIONS: FURTHER NOTES AND ILLUSTRATION

### E.1    MAX-MEAN APPROXIMATION

In equation 6, when approximating the reward distribution corresponding to the current posterior over Q-values, we suggest replacing the distribution of the state-value of the next state, i.e. the maximum of the Q-values for that state, by the distribution of the Q-value of the action with highest expected Q-value.

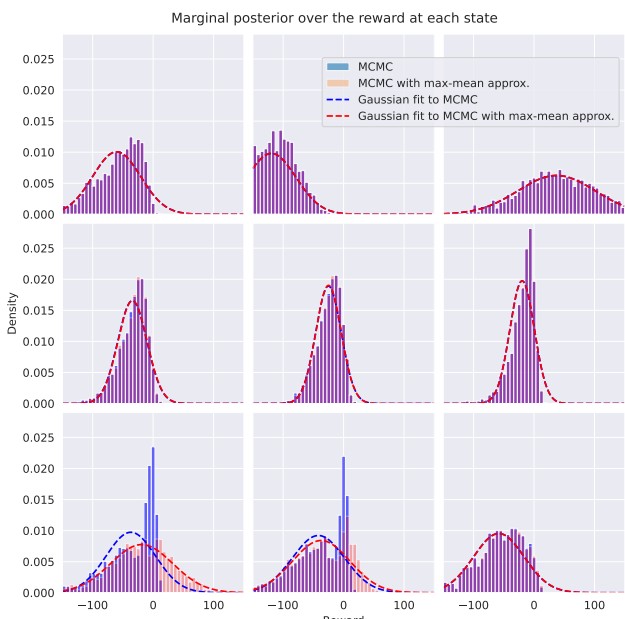

Figure 5: Effect of the approximation in Eq. equation 6 on the reward posterior. The plot shows the MCMC posterior distribution over rewards for each state, as well as samples deduced from the MCMC posterior over rewards deduced using the approximation from the posterior over state-values. Dashed lines show Gaussians fitted to the two MCMC samples. The results coincide except in the bottom left and middle squares.

.

**Reason for the approximation**  We are not aware of any good closed-form approximation for the mean and variance of the maximum of a general multi-variate Gaussian random variable (there exist approximations for the maximum of i.i.d. Gaussians, but here, the Q-values generally have different means, different variances, and can be arbitrarily correlated, all of which can strongly interfere with the i.i.d. approximation).

**Alternative approximation**  One could use sampling and the reparameterization trick instead. However, in the case of training, we observed the reparameterization trick destabilizing training and we did not succeed in matching the results of our approximation in the experimental settings presented in the main text.

**Illustration**  Figure 5 illustrates the effect of the approximation in the 3x3 gridworld introduced in the previous section. The approximation does not introduce any error in most states. This is because the posterior over Q values in the successor states to these states clearly favours a single action and thus the distribution of the state value conincides with the distribution of the highest-mean action. On the other hand, we can see the approximation introducing an error in two states at the bottom left. This is because the algorithm uncertain what the optimal action in the next state is, so the state-value does not coincide with the distribution of the max-mean Q-value. Consequently, the algorithm slightly underestimates the expected value of the next state and overestimates the reward.

### E.2 LIKELIHOOD APPROXIMATION

Following Lu et al. (2021), in Equation equation 9, we approximate the likelihood

$$p(a|s;\theta) = \int_{Q(s,\cdot)\in\mathbb{R}^{|\mathcal{A}|}} \frac{e^{\alpha Q(s,a)}}{\sum_{a'} e^{\alpha Q(s,a')}} q_\theta(Q(s,\cdot)) \, dQ(s,\cdot)$$

as

$$p(a|s;\theta) \approx \left( \sum_{a'} \exp\left( -\frac{\alpha\big(\mu_Q(s,a) - \mu_Q(s,a')\big)}{\sqrt{1 + 3\pi^{-2}\alpha^2(\sigma_Q(s,a)^2 + \sigma_Q(s,a')^2 - 2\Sigma_{aa'})}} \right) \right)^{-1}$$

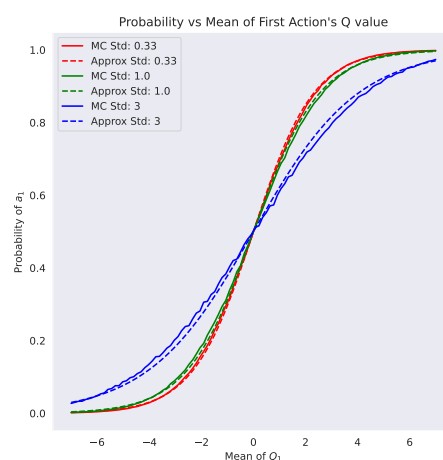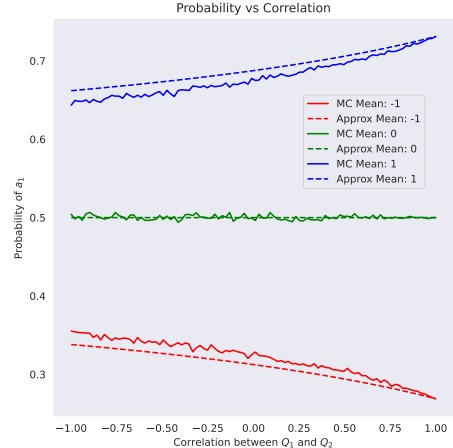

Figure 6: Comparison of the likelihood approximation equation 9 against a Monte Carlo evaluation using 10,000 Monte Carlo samples. Left: probability of $a_1$ as a function of the mean of $Q_1$ for 3 levels of standard deviation of $Q_1$. Correlation is set to 0 (the plot looks very similar for correlated values). Right: Probability of $a_1$ as a function of the correlation between $Q_1$ and $Q_2$ for three levels of the mean. Standard deviation is fixed to 1. Note the different scales on y-axes of the two plots.

.

.

To examine the quality of the approximation, we compare the probability of the action under the approximation against an approximation of the integral using Monte Carlo integration with 10,000 samples. In the test scenario, there are two actions, $a_1$ and $a_2$ with respective Q-values $Q_1$ and $Q_2$ that are jointly-normally distributed. We fix the marginal distribution of $Q_2$ to standard normal. We then try varying the mean and standard deviation of $Q_1$ as well as the correlation between $Q_1$ and $Q_2$.

Figure 6 shows the results. The approximation seems reasonably good, though it gets worse as correlation decreases.

Note that even with 10,000 samples, the Monte Carlo estimate displays notable noise. While we could use sampling and the reparameterization trick during training, we observed that this destabilizes training and we did not manage to get competitive results on any of the benchmarks. Thus we recommend using the proposed approximation that can be evaluated analytically.

### E.3 GAUSSIANITY

The Gaussianity assumption was illustrated in the previous example. Fig. 4 that for a Gaussian prior, the posterior stays reasonably close to Gaussian. The use of variational inference requires an assumption on the distribution family and the Gaussian is a natural starting point, on which further work can build to expand to other distribution families, especially if practical cases arise where the prior and associated posterior are notably non-Gaussian.

## F CODE AND DATA

All experiments were run using publicly available datasets and libraries. In particular, we used Python 3.10 (available under the GPL-compatible PSF license agreement; https://docs.python.org/3/license.html), Farama Gymansium 0.29.1 for the environments (MIT License), D4RL for the demonstration data (data used are available under the Creative Commons Attribution 4.0 License (CC BY), and code is licensed under the Apache 2.0 License), and PyTorch 2.0.1 (BSD-3 license).

We will release the full code on Github needed to replicate the experiments and result analysis when the anonymity requirement is lifted under a CC-BY license.

## G  COMPUTING RESOURCES USED

All experiments were run either on a single CPU (8-core AMD Ryzen 7 PRO 7840U, or 4 cores of a 64-core AMD Ryzen Threadripper 3990X) or a single NVIDIA RTX 3090 GPU, with roughly similar running times across these settings. A single learning run (i.e. running a single IRL learner on one set of demonstrations and then evaluating on the environment) took between 2 and 15 minutes. In total, less than 30 days of runtime on a single CPU or GPU as described were used for experiments across the whole development cycle.

