# OpenReview forum: "Q-based Variational Inverse Reinforcement Learning"
_ICLR.cc/2025/Conference — Submitted to ICLR 2025_

### Official Review · Reviewer_wYDB · 2024-10-21

**Soundness:** 2
**Presentation:** 2
**Contribution:** 2
**Rating:** 5
**Confidence:** 3

**Summary:**

This paper proposes a scalable Bayesian approach to inverse reinforcement learning (IRL) by learning a posterior distribution over Q-values instead of point reward estimates. This method enhances uncertainty quantification and robustness, particularly in safety-critical applications. QVIRL is demonstrated to be effective across various control problems and tasks.

**Strengths:**

QVIRL uses 1) variational inference for efficient learning, making it scalable to complex environments, 2) models uncertainty which provides robust, risk-averse policies by capturing uncertainty over rewards and 3) reduces computation time compared to MCMC-based IRL methods.

**Weaknesses:**

1. Sec 3 is a little bit stacked, so I recommend the authors provide a pseudo-code of their method for better clarity.
2. The algorithm described in Section 3 addresses discrete scenarios, while its extension to continuous actions relies on naive random sampling, which often fails to focus efficiently on relevant actions.

**Questions:**

1. Part of the legend in Figure 2 is missing. Also, the color in Fig 2 is too light to look clearly and the mean curve is missing. Also, where is random and expert?
2. There are also constraint inference works in safe-critical IRL, such as [1] and [2]. Why don’t the authors include them as baselines in evaluating QVIRL’s performance in Safety Gymnasium (Sec 4.3)?
3. The code is not provided, so I can not verify the experiment part.

[1] Hugessen, A., Wiltzer, H. and Berseth, G., 2024. Simplifying constraint inference with inverse reinforcement learning. In *First Reinforcement Learning Safety Workshop*.

[2] Malik, S., Anwar, U., Aghasi, A. and Ahmed, A., 2021, July. Inverse constrained reinforcement learning. In *International conference on machine learning* (pp. 7390-7399). PMLR.

---

> ### Author Response · Authors · 2024-11-22
>
> Thank you for the review. Regarding your questions:
>
> 1. We be happy to improve the figure, but would appreciate a few clarifications:
> > 1. Part of the legend in Figure 2 is missing.
>
> What part of the legend is missing? Do you mean that the legend is included only in the middle plot? If that is the case, would you prefer (the same) legend appearing within each of the three plots? Otherwise, it appears to us that all lines appearing in the plots are also included in the legend.
>
>
> > Also, the color in Fig 2 is too light to look clearly and the mean curve is missing.
>
> Regarding the colour being too light, are do you mean a particular colour or all of them? Or when printing the plot in black and white? When you say that the "mean curve is missing", does that mean that you would appreciate to see a curve showing mean performance across all models? Or is the mean for each method not displaying correctly for you?
>
>
> > Also, where is random and expert?
>
> The expert curve is a horizontal dashed line mostly coinciding with the curves of the other models (which is mentioned in the caption). In the right two plots, random performance is the dotted black line at the bottom (as shown in the legend). In the left plot, the random performance is at -500 but was cropped out to make the differences between the other methods easier to see. We have now added a note to the caption clarifying this.
>
> 2. There are also constraint inference works in safe-critical IRL, such as [1] and [2]. Why don’t the authors include them as baselines in evaluating QVIRL’s performance in Safety Gymnasium (Sec 4.3)?
> 	- [1],[2] use the framing of inverse constrained reinforcement learning (ICRL), which generally assumes that the primary (unconstrained) reward signal is known and only the safety cost is inferred. Our method, and the whole current evaluation setting, do not assume the knowledge of the primary reward. This is the reason why we did not include ICRL baselines.
> 	- That said, in line with the argument from [1], our method could be adapted to the ICRL setting by making it learn only the cost interpreted as a correction term subtracted from the primary reward. This would constitute a new evaluation setting.
> 	- Are you satisfied with the explanation of why the methods were not included (which we're happy to add to the paper), or would raising your evaluation score be conditional on us running the additional experiments on the ICRL setting? If it is important to you, I think there is a good chance we'd be able to get this done before the end of the discussion period - please let us know your stance for this.
>
> > 3. The code is not provided, so I can not verify the experiment part.
>
> We will provide a cleaned up version of the code upon deanonymized publication, including scripts for reproducing each experiment and notebooks for analysing results. If you deem this important, we should be able to provide a link to an unpolished version already now.
>
> Please let us know if we can offer any additional clarifications to these answers.

---

> > ### Comment · Reviewer_wYDB · 2024-11-25
> > **Thank you for the rebuttal**
> >
> > Thank you for the authors' response, which addressed some of my concerns.
> >
> > I think the problem regarding legend and color arises from my local PDF editor. Since the authors have added a note to the caption clarifying the random performance, I have no further questions about the plot.
> >
> > I recognize that the ICRL setting assumes a known nominal reward function, and I think evaluations with such a prior on whether and how much the performance would be elevated do contribute to the field and can be an enhancement for your experiment part.
> >
> > After considering the comments from other reviewers, my overall stance remains slightly negative.

---

### Official Review · Reviewer_rSkn · 2024-11-03

**Soundness:** 4
**Presentation:** 4
**Contribution:** 3
**Rating:** 8
**Confidence:** 4

**Summary:**

In this work, the authors provide a novel method for performing Bayesian Inverse RL which learns a Bayesian posterior over the Q values and derives a Bayesian posterior over rewards using the inverse Bellman equation. This improves over prior work (AVRIL) by learning a posterior over Q-values in addition to rewards, allowing the learnt posterior to be directly used to induce risk averse apprentice policies. Primarily, this is enabled by using the mean-field approximation proposed by Lu et al. to approximate the data likelihood. The authors show that their method is competitive with prior work in non-risk averse settings across multiple environment domains but with lower computational requirements. Importantly, the authors also demonstrate how the posterior over Q-values can be used to directly derive a risk averse policy that significantly outperforms prior work in terms of constraint violations on benchmark safety-constrained tasks.

**Strengths:**

- The paper is well written, easy to follow and understand. The derivations are clear
- The work is well-motivated - there is a clear need for better Bayesian IRL methods
- The authors evaluate their method across multiple environment domains and compare to several different baseline algorithms
- The authors evaluate their method over 10 seeds and (mostly) include errors bars/confidence intervals on the results
- The results are quite convincing, particulalry in the safety domain.
- The authors include an extensive limitations discussion

**Weaknesses:**

- In my opinion, it should be made more clear that the method uses an expert and non-expert (unlabeled) dataset. It is discussed in the Appendix that an additional unlabeled dataset of non-expert trajectories is used to estimate Eq. 6 in all experiments. Moreoever, it is not clear, which, if any of the baselines use an additional unlabeled dataset. The incorporation of an unlabeled dataset with transitions outside of the expert state distribution could substaintially improve online by providing the policy with additional knowledge of the dynamics of the MDP. It would preferable to compare to basleines that also incorporate this offline dataset (if the current baselines do not). In either case, it should be noted more prominently that this additional dataset is used in the experiments.
- I think it would be better if you could fit the discussion of extension to the case of unknown dynamics into the main paper. This is of primary interest since the prior methods you are comparing to all operate in the case of unknown dynamics and this is the typical setting for IRL.
- There are no confidence intervals in Table 2.

**Questions:**

- In sec 4.3 you state that: "These experiments were run in the online setting, where additional
auxilliary trajectories for evaluating the KL term are sampled from the apprentice policy throughout
the training. The baselines – BC, Inverse Soft Q-Learning (IQ; Garg et al. 2021), and AVRIL – were
run in the usual imitation learning setting."
Does this mean that you are comparing your method which is allowed online interactions with purely offline methods? Can you clarify what the "usual imitation learning setting" is?

- It would be interesting to see how your results in the safety domains change with different numbers of demonstrations. In theory regular IRL methods should also avoid the constraints given enough demonstrations since the expert trajectories are non-violating. Presumably the advantage of your method is that the constraints can be avoided with much fewer trajectories - but how many fewer?

---

> ### Author Response · Authors · 2024-11-22
>
> Thank you for your review and the time invested into reading and evaluating our paper.
>
> Response to the first listed weakness and first question:
> We tried to demonstrate the performance of the algorithm across 3 regimes:
> 1. fully offline, using only the expert demonstrations — this approach was applied to the Cartpole, Acrobot, and Lunar Lander environments to match the setting of the closest baseline, AVRIL.
> 2. additional offline transition (non-expert) data on the Mujoco environments (against matching baselines using the same assumption)
> 3. online (where the auxiliary transition data are generated during training) on the Safety Gymnasium. In this last setting (to which your question points), BC and AVRIL were run offline. IQ was run in an online setting. Sorry, the original phrasing was unfortunate. The comparison to offline methods can be considered somewhat unfair, though we included BC as a common simplest baseline for imitation learning (which is, in its basic form, fundamentally restricted to only the expert data). AVRIL was introduced as an offline method, but we still included (in its original offline form, as described and implemented by the authors) it as the closest Bayesian IRL method to ours. To provide a fairer comparison with this closest baseline, we have added a description of an online extension of AVRIL to Appendix D and aim to present the corresponding results by the end of the discussion period.
>
> Regarding the second question:
> Yes, we are confident that your presumption is correct and eventually, other imitation learning methods would close the gap with an increasing number of demonstrations. We will run the experiments and attempt to provide at least partial results before the end of the discussion period.
>
> Regarding the second listed weakness: thank you for the feedback. We will try to move more of the discussion into the main paper. By default, we'd do this by shortening the section on limitations and assumptions, though we are open to suggestions for other candidates for cutting or moving to the appendix.
>
> Regarding **confidence intervals**, we omitted them to declutter the table and improve readability, but we will include the full table with confidence intervals in the appendix before the discussion period ends.
>
> We appreciate your feedback and are happy to provide additional details or address further questions, if you have any.

---

> > ### Comment · Reviewer_rSkn · 2024-11-25
> >
> > Thank you for your response and the additional clarifications.
> >
> > Regarding the three settings, as I understand it, you need to estimate Eq. 6 from data, so in case 1. the fully offline case is doing this only on the expert data. This will of course be very brittle, as you note yourselves in the Appendix. and is probably quite restricted to the simple environments that you evaluate in this section. For this reason, I think it is much more appropriate to frame your method as one that requires some unlabeled data (offline or online) and push the fully offline experiments to the Appendix. Indeed, you say yourselves in the appendix "We think our method is more useful in settings
> > where some additional source of information about the environment dynamics is present," - I believe it should be presented as such throughout the main body of the paper. This would also include ensuring the appropriate baselines re the discussion of comparing fully offline to online. Thank you for the addition of dynamics-aware Avril method in the appendix - I look forward to seeing the corresponding results.
> >
> > Moving the fully offline section to the Appendix would also give more room for the additional experiments on the expert data size. I think this is very important since the primary contribution of your work is the ability to get uncertainty estimates on Q-vlaues and hence risk averse policies, which you agree is primarily an advantage in the low data domain.

---

> > > ### Author Response · Authors · 2024-11-26
> > >
> > > Thank you. Yes, we agree with the point about our method being suited to settings with additional dynamics information, unless expert data provide sufficiently dense coverage of all relevant parts of the state-action space. We are happy to change the emphasis of the paper in that direction and move the offline setting to the appendix (the main argument for the inclusion was that this was the setting that the closest prior method, AVRIL, adopted, and evaluating on a setting matching this prior work seemed appropriate for fair comparison).
> > >
> > > We will try to get back with the data size results soon. Also, we'll try to add an evaluation of the value of adding different amounts of additional dynamics information to the final version of the paper, but don't expect to have this by the discussion period.

---

> > > > ### Author Response · Authors · 2024-12-04
> > > >
> > > > We have added the results for online AVRIL to the paper as well as the confidence intervals into the appendix. (see the updated pdf)
> > > >
> > > > We did not get the results for data scaling in time for updating the pdf but have preliminary results now. We have varied the number of expert demonstration trajectories from 1 to 100 (1, 5, 10, 20, 50, 100) and ran BC, online AVRIL, IQ-Learn, and QVIRL on each setup. So far, preliminary takeaways regarding safety:
> > > > - for BC, even 100 demonstrations are generally not enough to make it behave safely
> > > > - online AVRIL generally needs about twice as many trajectories as risk-averse QVIRL to reach a similar level of safety, with IQ-Learn trailing slightly behind
> > > > - on the other hand, IQ-Learn tends to outperform AVRIL in terms of primary reward and often slightly outperforms QVIRL
> > > > We will add full results into the next version of the paper.

---

### Official Review · Reviewer_WkWT · 2024-11-03

**Soundness:** 3
**Presentation:** 3
**Contribution:** 3
**Rating:** 5
**Confidence:** 4

**Summary:**

This paper introduces a new method for scalable Bayesian inverse RL /
apprenticeship learning, where the posterior over reward functions is
approximated by variational inference over a class parameterized by
neural networks. Exploiting a bijection between Q-values and rewards
under a particular rationality model, the proposed QVIRL method can
maintain posteriors over Q-values, eliminating the need for a policy
optimization subroutine. Moreover, using the posterior over
action-values, the authors argue for (and experiment with) optimizing a
risk-averse utility during apprenticeship learning, to stay closer to
the demonstration data for the purpose of recovering a safe policy.
Through experiments on a variety of benchmarks, the authors shows that
QVIRL is often quite performant, often exceding the performance of all
baselines.

**Strengths:**

This paper presents a fresh approach to solving Bayesian IRL and
apprenticeship learning. I really appreciated the extent to which the
authors leveraged the Bayesian perspective—particularly with regard to
the posterior variance of inferred Q-values—to improve the safety of the
resulting apprentice policy. The empirical results appear to be quite
good. Moreover, the paper is expertly written—it was very easy to follow
and enjoyable to read.

**Weaknesses:**

The main weakness of the paper to me is that the consequences of the
various approximations made are not discussed explicitly or in detail.
For instance, under what circumstances would the mean-field
approximation of the likelihood cause problems?

Moreover, the reward prior does not appear to be discussed explicitly; I
see no mention of which prior was used in the experiments. There appears
to also be an assumption made on the reward prior in equation (10), that
is the priors over rewards for each state-action pair are independent.

Finally, it is not clear how many seeds were used for evaluation in the
deep RL tasks. In fact, it reads as though only one seed was used, which
makes it difficult to assess whether the performance improvements
presented are statistically significant.

**Questions:**

On line 226, what do you mean by a "variational distribution"?

Can you explain in further detail what the consequences are of the
approximation in equation (6)? Maybe a simple figure would be nice
(comparing say the density of the posterior of $\max Q$ vs that of
$Q(\cdot, \arg\max)$). Is this a novel contribution?

Why are there no confidence intervals for the SafetyGym results?

How many seeds are used in the experiments?

---

> ### Author Response · Authors · 2024-11-22
>
> Thank your for your review. Here are answers to your questions:
>
> > 1. On line 226, what do you mean by a "variational distribution"?
>
> We mean the parameterized family of distributions whose parameters variational inference is learning. That means that the model is assuming that the real posterior over Q-values can be well approximated by a distribution from this family (in our experiments, we assume Gaussian).
>
> > 2. Can you explain in further detail what the consequences are of the approximation in equation (6)? Maybe a simple figure would be nice (comparing say the density of the posterior of vs that of ). Is this a novel contribution?
>
>  In the approximation, we are replacing the maximum of |A| random variables by the random variable with the highest mean. This approximation is good if (1) there is an action that has a high probability of being optimal (i.e. one of the Q values has a mean higher than the others by a margin that is large relative to the variance of the Q values), or (2) the action values are closely correlated and have similar variance. If there are many independent actions with high variance, the approximation underestimates the value contribution of the next state, which, given a prior over rewards, may underestimate the Q-value of the given state-action pair. This has the nice property that this discourages taking actions for which the approximation may be crude, which can be seen as a kind of self correcting mechanism.
>
> We will try to produce a visualization by the end of the discussion period.
>
> Yes, we'd consider this novel in this setting, and agree that in light of this, it would deserve deeper discussion, which we'll try to add to the paper.
>
> > 3. Why are there no confidence intervals for the SafetyGym results?
>
> There are two numbers for each model as is, and the table was quite cluttered when confidence intervals were included so we removed them from the main-text table for easier readability. We agree the confidence intervals should appear in the paper, and we will add them to the appendix before the end of the discussion period.
>
> > 4. How many seeds are used in the experiments?
>
> 10 random seeds per experiment. In general, each seed was also run with a different set of expert demonstrations. (We write this in Appendix C.2, l. 885)

---

> > ### Comment · Reviewer_WkWT · 2024-11-26
> >
> > Thanks to the authors for their response. I appreciate the clarifications about the experimental setup. However two of my weaknesses are unaddressed in your response. I believe the first weakness I pointed out was partly discussed in the response to 2Mij. Do you have any comments about the second weakness that I raised?

---

> > > ### Author Response · Authors · 2024-11-26
> > >
> > > Apologies for a delay in the response to the weaknesses - we were working on incorporating those into the paper and it has been taking longer than expected. The updated paper now contains Appendix E where we started assessing the various approximations - you can let us know, if this goes in the direction you were looking for. (This is still work in progress and we will try to add comments on all approximations at least to the level of E.1 and will try to explain the reason for each approximation, possible alternatives and evaluate its fidelity.)
> > >
> > > Regarding the second weakness - yes equation (10) currently assumes independence and as we noted already in the original manuscript, Appendix C.3, we were using a standard normal prior for each state action pair (except for Lunar Lander where we used a standard deviation of 5). We chose the independent prior mainly to match AVRIL's implementation and thus keep the assumptions of the two methods close.
> > > However, it is true that it would be more natural to use e.g. a Gaussian process prior, and we definitely should have described in more detail how to incorporate such a prior. Let us give an outline here:
> > > Note that after linearization in Eq. (6), we can take any subset of state-action pairs, and since the vector of rewards on these state-action pairs is a linear function of the appropriate Q-values, we can easily deduce the covariance matrix of the rewards from the covariance of the Q-values. Then Eq. (10) can calculate the KL divergence between this joint distribution over rewards and the (joint) prior (i.e. the multivariate Gaussian obtained by evaluating the Gaussian process prior on the same state-action pairs).
> > > This process can be done over all state-action pairs in small tabular settings, or it can be done on a per-batch basis if we need to sub-sample.
> > > We will expand this answer and add it to the paper.

---

> > > > ### Comment · Reviewer_WkWT · 2024-11-28
> > > >
> > > > Thanks for the clarifications.
> > > >
> > > > Firstly, please add references to these appendix sections in the main body of the text where the corresponding assumptions are implicitly made. These are not simply 'implementation details', since the method does specifically require certain assumptions on the priors.
> > > >
> > > > Second, Appendix E is a very important section---this should definitely be in the main text, since it involves yet another approximation that you defined. I am also not particularly convinced about this approximation; I think it is sensible to some extent, but it's unclear to me why, given that you're making such an approximation, you need Gaussian priors/posteriors. Also, is there any way to use some maximum entropy distribution to deal with representing the distribution over the maximum of Q-value posteriors?
> > > >
> > > > Finally, while not super important, I'm curious if the independent normal prior _per state-action pair_ introduced any difficulties, since this admits reward function samples with very low regularity. You propose using GP priors as a substitute for future work, which is sensible and would solve this issue. Why didn't you try this in your experiments? Is it because it would be difficult to scale as you collect data?

---

> > > > > ### Author Response · Authors · 2024-12-04
> > > > >
> > > > > Thank you for your response. We have added the reference to the discussion of approximations to the main text (after the pdf update deadline). Appendix E does not introduce any new approximations - it just explains their effects in more detail, though we will try to add a short summary of this into the main text after each approximation is introduced.
> > > > > We have spent considerable time looking for a suitable approximation of the distribution of the maximum and have not found any that would fit this use case. Even the case of the maximum of 2 jointly normal random variables is subject to ongoing research. There are decent approximations for the maximum of i.i.d. Gaussians, but unfortunately, this does not seem very useful here.
> > > > >
> > > > > Regarding the GP priors: we found that the neural network encoder, especially in conjunction with L2 regularization, induces an implicit smoothness prior that we found to work well in practice. Also, in general, we were wary to introduce additional assumptions that prior work did not use, especially since GP priors require additional hyperparameters.
> > > > > That said, we will re-run the experiments with a GP prior as we would say it is the proper way of doing things. This comes at the cost of added computational complexity of handling the full covariance matrix across a whole training batch of state-action pairs (with some operations cubic with respect to the batch size).

---

### Official Review · Reviewer_2Mij · 2024-11-03

**Soundness:** 1
**Presentation:** 2
**Contribution:** 1
**Rating:** 3
**Confidence:** 3

**Summary:**

This paper addresses scalability and posterior uncertainty estimation in Bayesian inverse Reinforcement Learning (RL) through Q-values variational inference for approximating reward posterior distributions. The methodology largely builds upon two existing works: (1) VAE [1]:  Adopts the concept of variational posterior yielding joint multivariate Gaussian distributions as a main standing point over this work, as shown in Figure 1, and the two posterior distributions are also assumed to be approximated well by Gaussian; (2) AVRIL [2]: the overall structure and key definitions are similar, from "apprenticeship learning", "learning a variational posterior over reward" to "Boltzmann distribution" in this paper. The authors compare their approach against both Bayesian inverse RL methods like [2] and state-of-the-art imitation learning approaches like [3]. However, there are significant concerns regarding both methodological innovation and experimental demonstration.

**Strengths:**

1. The idea and algorithmic structure including key definitions are foundational on existing literature, e.g., [1] and [2], which makes the overall structure of the paper generally easy to follow.

2. The paper adeptly integrates a multitude of concepts such as Bayesian Inverse reinforcement learning, apprenticeship learning, variational inference, and uncertainty estimation over Q-values. These topics have been at the forefront of recent research trends.

3. The paper is motivated to address safety-critical applications with a specific focus on scalability and uncertainty estimation, though the scalability claims lack evident support.

**Weaknesses:**

1. The paper has limited novelty: the central idea is heavily derived from existing work, particularly VAE [1] and AVRIL [2]. Although the authors attempt to distinguish their approach, such as through "uncertainty estimation over Q-values instead of a point estimate" (page 4), the paper lacks substantial novelty, with much of its contribution reflecting [2]. For instance, in Table 2, the reward outcomes show minimal improvement over baseline methods.

2. The work builds on established algorithms but does not address or resolve inherent limitations within them. For example, it assumes Gaussian approximations for the posterior distributions over Q-values and rewards—a limitation acknowledged in both [1] and [2]. Relying on Gaussian assumptions may restrict model accuracy since Gaussian distributions cannot capture all characteristics of arbitrary distributions.

    Furthermore, the paper employs a maximum entropy approach, which is not a true distance metric. More effective and precise metrics, such as those used in optimal transport, could enhance the results.

3. The contribution of "scalability" is weak. The paper does not provide theoretical support for scalability, and the little empirical evidence on page 8 is similarly insufficient.

4. Several typos - e.g., "finite state- and action ..." (page 2) and "the two closest prior methods for Bayesian inverse reinforcement." (page 7).

**Questions:**

1. What specific aspects of the approach contribute to improved scalability? And, How is scalability quantitatively demonstrated in experiments (and compared to AVRIL [2])? It appears that the proposed method does not improve on [2] in this regard.

2. In Equation 2, it would be great to justify the construction of likelihood p(r|D) over the Boltzmann rationality model in Equation 2.

3. How is the accuracy of the Gaussian-based posterior estimation ensured? Can the authors propose a metric or analysis to support this assumption? Given that the variational distribution is approximated as Gaussian, this question is critical when using variational inference.

__Reference__:

[1] Kingma, Diederik P. "Auto-encoding variational bayes." arXiv preprint arXiv:1312.6114 (2013).

[2] Chan, Alex J., and M. van der Schaar. "Scalable Bayesian Inverse Reinforcement Learning." International Conference on Learning Representations. 2021.

[3] Jarrett, Daniel, Ioana Bica, and Mihaela van der Schaar. "Strictly batch imitation learning by energy-based distribution matching." Advances in Neural Information Processing Systems 33 (2020): 7354-7365.

---

> ### Author Response · Authors · 2024-11-22
>
> Thank you for your time and effort invested in reading our paper and providing feedback.  Here are our responses to some of your concerns as well as your questions:
>
> Regarding **novelty and contribution** (Weakness 1, part of 2): Our key contribution is achieving meaningful uncertainty quantification in Q-values and resulting policies while maintaining the computational efficiency of methods like AVRIL (which, we argue, does not provide useful uncertainty quantification). This is not a trivial extension - it requires making the Q-value distribution consistent with both the reward prior and Bellman equations while tracking covariance across the state-action space; this is a large part of the paper and something not achieved by AVRIL. The practical value of this contribution is demonstrated through:
> 1. The ability to produce risk-averse policies, which is not possible with AVRIL's point estimates (as demonstrated in our Safety Gymnasium experiments)
> 2. Significant and consistent performance improvements across tasks even in standard (non-risk-averse) settings, as shown in our experimental results.
> 3. Newly added analysis in Appendix D showing and explaining that AVRIL's uncertainty quantification abilities are inadequate.
>
> Regarding **scalability** (Weakness 3, Question 1): We would like to clarify that our goal was not to improve upon AVRIL's scalability, but rather to achieve meaningful uncertainty quantification while maintaining similar computational efficiency. The significant scalability improvement we highlight is relative to MCMC-based methods that offer good uncertainty quantification but take hours to run on classic control environments where our method takes minutes (lines 399-402). Furthermore, we apply our method to higher dimensional problems than any prior work in Bayesian IRL, demonstrating practical scalability while preserving uncertainty quantification (though yes, AVRIL scales similarly to our method).
>
> **Boltzmann rational likelihood** (Question 2): The desirable properties of the Boltzmann rationality model include the fact that unlike a lot of work in IRL, it does not assume perfect optimality of the expert. We assume that if two actions are both similarly close to optimal, they're similarly likely to be taken. As the immediate regret associated with an action grows, that action gets exponentially less likely. Building on this, equation (2) expresses that conditional on the Q value, actions taken by the expert are independent - i.e. their probability depends only on how "good" the action is (from the point of view of fulfilling the goal represented by the reward), rather than other factors (including choices made by the expert in other states). This is a standard assumption almost universal in IRL literature. Would it deserve further scrutiny? Absolutely. However, since our key contribution lies elsewhere (see above), we chose to leave this assumption fixed to better isolate the impact of our primary innovation.
>
> **Gaussianity assumption in QVIRL** (Question 3, Weakness 2): This is an important consideration. Unfortunately, the likelihood lacks a conjugate prior so we cannot guarantee a particular precise form of the posterior and we do need to resort to approximation. We agree that it is important to track whether a chosen approximation is appropriate for a particular IRL problem at hand.
> 1) This can be easily investigated in cases where we have access to a more exact way of estimating the true posterior. This can be done using MCMC methods that can simulate samples from the true posterior, though they are applicable only to small problems. We have done this for a number of gridworld problems and now attach one such example as Appendix D2, which we encourage you to see. In these cases, especially the variational posterior over Q-values, which is at the centre of our algorithm and is used to determine the apprentice policy, seems to match the true posterior well. We observe similar behaviour across other gridworlds (though all with a Gaussian prior).
> 2) Further empirical evidence for the adequacy of the Gaussian assumption seems to be the favourable empirical results even on continuous domains, especially the experiments with risk averse policies on Safety Gymnasium.
> 3) The method could also be adapted to non-Gaussian distributions. To account for multimodal distributions, a mixture of Gaussians could easily be used instead of a Gaussian. Transformations to other distributions could be achieved, e.g. using normalizing flows, though additional work may be required to adequately transform the likelihood and the associated approximation (Eq. 9), but we consider this feasible.
>
> Thank you for pointing out the 2 typos. We have fixed them and will again proofread the paper before uploading the next version.
>
> Please let us know if this answers your questions and addresses some of the concerns, or we can provide any further clarifications.

---

> > ### Comment · Reviewer_2Mij · 2024-11-25
> > **Thank you for the authors’ response**
> >
> > Thank you for the authors’ response and efforts. Regarding scalability, as noted in the response, it is similar to AVRIL, with the algorithm framework primarily inherited from AVRIL as well. Thus, I suggest removing scalability from one of the two claimed contributions (i.e., scalability and full posterior uncertainty estimation) in this paper.
> >
> > For Equation 2, it would be great if a justification or references that could justify the equation were provided, rather than relying on "This is a standard assumption almost universal in IRL literature". All assumptions in the paper should be reasonable, but this one seems problematic and warrants further clarification.
> >
> > I appreciate the response regarding the Gaussianity assumption. However, exploring optimal transport theory (which accommodates arbitrary distributions) might lead to a reassessment of the response. After considering the comments from other reviewers, my overall stance on the paper is still unfortunately negative.

---

> > > ### Author Response · Authors · 2024-11-26
> > >
> > > Regarding contribution: there exist two strands of prior work - that based on MCMC, which achieves good uncertainty quantification but is slow, and AVRIL which is faster but with very poor uncertainty quantification. QVIRL has both strengths, so we think it makes sense to emphasize both in the text. We will try to get back with an improved formulation that makes it clear that we're not improving scalability relative to AVRIL.
> > >
> > > Thank you for suggesting that a method based on optimal transport theory may be suited for the problem at hand. We are keen to explore that direction. However, many of the particulars would need to change and the result of that endevour, if successful, would be a different method altogether. Here, we have described a method based on KL-based variational inference and we believe we have demonstrated notable advantages it has over prior state-of-the-art. The Gaussian assumption seems to do an adequate job on the tasks at hand. It is almost certain that an even better method exists, but we think that is for future work to explore, and our paper still presents meaningful progress on the problem at hand. We are open to further suggestions how to better demonstrate that.

---

### Meta-Review · Area_Chair_kSHo · 2024-12-23

**Metareview:**

The paper describes a new Bayesian inverse RL technique called QVIRL that leverages variational inference.  It is claimed that QVIRL enables the estimation of distributions over Q-functions as well as being scalable.  The strengths of the paper are a valuable Bayesian IRL technique, mathematical depth in the derivation of the approach and good empirical results.  The weaknesses include a mismatch between the claim of scalability and the evidence provided, numerous approximations and lack of discussion and comparison with Bayesian inverse constraint learning techniques since QVIRL is tested on problems with constraints where the cost of violating those constraints must be estimated.  While QVIRL is promising, the paper is not ready for publication.

**Additional Comments On Reviewer Discussion:**

The discussion among the reviewers focused on the lack of discussion and comparison to Bayesian inverse constraint learning techniques.  At some level, learning constraints is orthogonal to learning rewards, but the paper emphasizes the need to learn the cost of violated constraints for safety purposes and reports results on constrained MDPs where the goal is to learn a policy that respects some underlying constraints.  Hence, inverse constraint learning techniques are alternatives that are more natural than IRL techniques since they directly learn the underlying constraints instead of assuming that costs can be learned for constraint violation.  See below for a list of papers on Bayesian inverse constraint learning.

Another topic of discussion was the claim of scalability.  Lines 87-88 claim that QVIRL scales to problems with high dimensional state spaces and high number of demonstrations.  The experiment section reports that QVIRL trains in minutes compared to hours for previous techniques.  This is great, but the dimensionality of the state space for those problems seem to be low and the amount of demonstration data is not clear. A graph that shows the running time of QVIRL as we vary the amount of data is missing.  Similarly a graph that shows the scalability of QVIRL as we test with problems of increasing state space dimensionality is also missing.   There is no theoretical analysis either of the scalability of QVIRL with respect to state dimensions and amount of demonstrations.

Finally, there was also some discussion about the Gaussian assumption and the various approximations used by QVIRL.  The reviewers appreciate that the assumption and the approximations are needed to ensure scalability, but they obviously introduce a tradeoff since there must be an impact on the accuracy of the posterior.  Some discussion and evaluation of the impact on accuracy is missing.  For instance, measuring the expected calibration error due to the assumption and approximations is needed.

Bayesian inverse constraint learning papers:

Glazier, A., Loreggia, A., Mattei, N., Rahgooy, T., Rossi, F., & Venable, K. B. (2021). Making human-like trade-offs in constrained environments by learning from demonstrations. arXiv preprint arXiv:2109.11018.

Papadimitriou, D., Anwar, U., & Brown, D. S. (2022) Bayesian Methods for Constraint Inference in Reinforcement Learning. Transactions on Machine Learning Research.

Liu, G., Luo, Y., Gaurav, A., Rezaee, K., & Poupart, P. (2023) Benchmarking Constraint Inference in Inverse Reinforcement Learning. In The Eleventh International Conference on Learning Representations.

Xu, S., & Liu, G. (2023). Uncertainty-aware Constraint Inference in Inverse Constrained Reinforcement Learning. In The Twelfth International Conference on Learning Representations.

Subramanian, S. G., Liu, G., Elmahgiubi, M., Rezaee, K., & Poupart, P. (2024). Confidence aware inverse constrained reinforcement learning. ICML.

---

### Decision · Program_Chairs · 2025-01-22

Reject